# Could We Expect Postoperative Cup Anteversion after Total Hip Arthroplasty Using Postoperative Plain Anteroposterior and Lateral Radiograph? A Three-Dimensional Experimental Operation Study

**DOI:** 10.3390/jcm12206664

**Published:** 2023-10-21

**Authors:** Tae Sung Lee, Hyuck Min Kwon, Jun Young Park, Dong Ki Kim, Kyoung Tak Kang, Kwan Kyu Park

**Affiliations:** 1Department of Orthopedic Surgery, Severance Hospital, Yonsei University College of Medicine, Seoul 03722, Republic of Korea; skisports88@yuhs.ac (T.S.L.); hyuck7777@yuhs.ac (H.M.K.); jjunyon@yuhs.ac (J.Y.P.); gabrian1201@yuhs.ac (D.K.K.); 2Department of Mechanical Engineering, Yonsei University, Seoul 03722, Republic of Korea

**Keywords:** total hip arthroplasty, dislocation, anteversion, inclination

## Abstract

Background: A postoperative radiograph in total hip arthroplasty (THA) is usually obtained to evaluate the inclination and anteversion of the acetabular components. However, there is no gold-standard method for calculating the exact inclination and anteversion of the acetabular components on post-THA radiographs. We aimed to measure the actual anteversion of the acetabular component on postoperative radiographs by obtaining correlation data between the virtual and actual acetabular component positioning using virtual three-dimensional (3D) surgery. Methods: A total of 64 hip scans of 32 patients who underwent lower-extremity computed tomography (CT) were retrospectively reviewed. We reconstructed 3D models of the 64 hips using customized computer software (Mimics). Furthermore, to identify the safe zone of acetabular component position in THA, we performed virtual 3D surgery simulations for five anteversion (−10°, 0°, 10°, 20°, and 30°) and five inclination (20°, 30°, 40°, 50°, and 60°) types. We analyzed the acetabular anatomy using 3D models to measure the radiographic, anatomical, and operative anteversion (RA, AA, OA) and inclination (RI, AI, OI) angles. Additionally, we used the Woo–Morrey (WM) method to calculate the anteversion angle in the reconstructed cross-table lateral (CL) radiographs and determined the correlation between these measurements. Results: The safe zone of the acetabular component was visualized on post-THA CL radiographs using the WM method of anteversion measurement based on the different anteversions and inclinations of the acetabular component. The AA, RA, OA, OI, and WM differed significantly between males and females (*p* value < 0.05). As the anatomical inclination or anteversion increased, the WM anteversion measurements also increased. The radiographic anteversion measurement best matched the WM method of measurement, followed by anatomical and operative methods. Conclusions: The actual anteversion of the acetabular component after THA can be measured on CL radiographs with the WM method using a 3D virtual program, with good reproducibility.

## 1. Background

According to Statistics Korea, 31,301 cases of hip reconstruction surgery, including total hip arthroplasty (THA), were performed in 2020 [1]. Since John Charnley pioneered hip replacement surgery in 1960 [2], the demand for THA in the treatment of arthritis, femoral head osteonecrosis, and hip developmental dysplasia has been increasing annually. However, dislocation after THA is a devastating complication [3,4,5,6], with an incidence of 1.9–2.3% [3,7]. To reduce dislocation after primary THA, dual mobility acetabular articulation was introduced in the 1970s and showed beneficial outcomes for reducing the dislocation risk in patients with a higher risk of instability [8]. Although the incidence rate is low, once it has occurred, it can lead to revision surgery [9,10]. The risk factors of dislocation after THA are divided into patient-related and surgical risk factors [4]. Patient-related risk factors include neuromuscular and cognitive disorders, cerebral palsy, muscular dystrophy, psychosis, dementia, and alcohol consumption [11]. Additionally, surgical risk factors include the surgical approach, soft-tissue tension, component positioning (cup anteversion and inclination), impingement, cup size, head size, liner profile, and surgeon experience [4]. These factors, particularly the positioning of the acetabular component, can be adjusted to reduce the number of dislocations [12]. If, in the case of the posterior approach of THA, the anteversion of the acetabular component is not enough, there could be a tendency to dislocate posteriorly.

During the planning and execution of surgery, several intraoperative factors are carefully considered by surgeons, including the adequate positioning and orientation of the components [5,13]. Lewinnek et al. described a “safe-zone” of cup positioning: 5–25° of anteversion and 30–50° of inclination [14]. Inappropriate anteversion and inclination can cause impingement and levering out, which can lead to dislocation [15,16]. Therefore, a post-THA radiograph is usually obtained for assessing the anteversion and inclination of the acetabular component. Although the inclination of the acetabular component can be easily measured on plain anteroposterior (AP) radiographs, anteversion is difficult to measure. Several studies have focused on evaluating the anteversion of the acetabular component on plain AP radiographs [14,17,18,19,20]. However, no gold standard method has been established to date, making it difficult to determine the exact anteversion on plain AP radiographs [21,22]. Several studies have also focused on plain cross-table lateral (CL) radiographs (Figure 1) for evaluating the anteversion [23,24,25,26]; they provide an acceptable assessment of the general component position [23,24,27,28]. In a previous study, the anteversion of the acetabular component was measured using the method described by Woo and Morrey (WM) [5] and the ischiolateral method described by Pulos et al. [23]. The anteversion of the acetabular component measured using these plain radiographic methods was consistently valid, with good interobserver reproducibility [29]. However, these methods do not show the actual anteversion, which can be precisely measured using computed tomography (CT). Murray defined and measured the acetabular orientation using three methods (radiographic, anatomical, and operative measurements of inclination and anteversion) [28]. Thus, the terms “inclination and anteversion” are frequently confused with these three methods. Inclination is usually defined as radiologic inclination, which can be measured in the coronal plane on a CT scan. Anteversion is usually defined as anatomical anteversion, which can be measured in the axial plane on a CT scan. However, according to Murray et al. [28], there are three inclinations and anteversions.

Currently, there is no gold-standard method for calculating the exact inclination and anteversion of the acetabular components in post-THA radiographs. Surgeons often obtain only a postoperative radiograph, and not a CT, due to the effects of radiation and to ensure cost-effectiveness. Furthermore, only a few studies have focused on the measurement of acetabular orientation. Therefore, in this study, we aimed to determine the actual anteversion of the acetabular component on postoperative radiographs by obtaining correlation data between virtual and actual acetabular component positioning using three-dimensional (3D) virtual surgery. We also aimed to analyze the acetabular anatomy to calculate the radiographic, anatomical, and operative anteversion and inclination, to determine the correlation between them.

## 2. Methods

### 2.1. Data Collection

This study was approved by the Institutional Review Board. A total of 32 patients (16 males and 16 females) who underwent THA between January 2011 and May 2022 were included in this study, and their data were retrospectively reviewed. Patients who had been diagnosed with osteoarthritis or osteonecrosis of the femoral head and had undergone primary THA were included. Patients who had been diagnosed with septic arthritis, had undergone previous surgery on the hip and spine, had a dysplastic hip, or had sequelae of childhood hip diseases were excluded. Preoperative CT data were saved in the digital image communication in medicine (DICOM) format.

Using Mimics (version 19.0; Materialize, Leuven, Belgium), a 3D reconstruction of the pelvic bone was performed. Furthermore, by setting the hip center in the acetabulum, half of the acetabular component hemisphere was fixed. Thereafter, to identify the safe zone of the acetabular component position in THA, we performed virtual 3D surgery simulations for five anteversion types (−10°, 0°, 10°, 20°, and 30°) and five inclination types (20°, 30°, 40°, 50°, and 60°).

We obtained plain CL radiographs using the method commonly used in practice. The radiation beam was directed through the groin, with the opposite lower extremity excluded from the imaging field. A cassette was positioned on the side of the hip at a right angle relative to the incidence angle; thus, it was projected toward the groin region at an angle of 45°, parallel to the longitudinal axis of the pelvis [30]. A 3D reconstruction image was rotated 45° internally and then projected onto a 2D plane to generate a plain CL radiograph (Figure 2). Acetabular anteversion was calculated using the WM method, which is computed using the angle between a line perpendicular to the horizontal plane of the film and a line along the opening of the acetabular component [29] (Figure 2).

The coronal plane was defined as the plane connecting three points: the bilateral anterior superior iliac spine (ASIS) and the center of the symphysis pubis. The sagittal plane was defined as the plane connecting the center of the symphysis pubis and the sacral crest. The transverse plane was automatically defined by the two other planes (coronal and sagittal).

Furthermore, the orientation of inclination and anteversion was calculated using three methods (radiographic, anatomical, and operative measurements). Anatomical anteversion (AA) was defined as the angle between the transverse and acetabular axes when it is projected onto the transverse plane. The anatomical inclination (AI) was defined as the angle between the acetabular and longitudinal axes. Operative anteversion (OA) was defined as the angle between the longitudinal axis of the patient and the acetabular axis projected onto the sagittal plane. Operative inclination (OI) was defined as the angle between the acetabular and sagittal planes. Radiographic anteversion (RA) was defined as the angle between the acetabular and coronal planes. Radiographic inclination (RI) was defined as the angle between the longitudinal and acetabular axes when projected onto the coronal plane (Figure 3) [28].

Thereafter, to identify the safe zone of the acetabular component position in THA, we performed virtual 3D surgery simulations for five anteversion types (−10°, 0°, 10°, 20°, and 30°) and five inclination types (20°, 30°, 40°, 50°, and 60°). Subsequently, 64 hip CT scans of 32 patients were analyzed. The acetabular component was set in a normal position based on the positioning of the patient’s own acetabulum. Subsequently, we analyzed the hip acetabular anatomy to calculate the radiographic, anatomical, and operative anteversion and inclination angles to determine the correlation between them.

### 2.2. Statistical Analysis

To obtain a power of 0.95 (1 − β) with an α of 0.05, the calculated sample size was 27 cases per group [31,32]. Considering a dropout rate of 20%, the target sample size was 32 cases per group. The Shapiro–Wilk test was performed to determine the normal distributions for anteversion and inclination angles. Pearson correlation and regression tests were performed to evaluate the correlation between the three methods and the WM method of anteversion measurement. When the correlation coefficient is closer to 1, there is a strong correlation. When the correlation coefficient is closer to 0, there is a weak correlation. Statistical analyses were performed using IBM SPSS Statistics for Windows (Version 25.0; IBM Corp., Armonk, NY, USA), and *p*-values < 0.05 were considered significant.

## 3. Results

Inclinations and anteversions in each of the three methods of measurement, including sex distribution, were measured. The AA, RA, OA, OI, and WM anteversions differed significantly between males and females (Table 1). The mean values of AA, RA, OA, and WM anteversion were 20.1 ± 6.1, 15.5 ± 4.8, 23.3 ± 7.3, and 15.6 ± 4.8, respectively. Furthermore, the mean values of AI, RI, and OI were 51.5 ± 6.1, 49.5 ± 4.9, and 46.9 ± 4.5, respectively.

Figure 4 shows a plain post-THA CL radiograph and the WM anteversion measurement according to different anteversions and inclinations of the acetabular component. For example, if the inclination of the acetabular component was 30° and the WM method of anteversion was 10.3°, the actual anteversion of the acetabular component was 20°. If the inclination of the acetabular component was 0°, then the WM method of anteversion was also 0°. In the final schematic, the inclination was measured using the WM method. As the anatomical inclination or anteversion increased, the WM anteversion measurements also increased.

All three anteversion measurement methods (AA, RA, and OA) showed strong significant positive correlations with the WM method of measurement. Radiographic measurements best matched the WM method (correlation coefficient: 0.999), followed by anatomical and operative measurement methods (0.972 and 0.957, respectively). Using a regression analysis, all three anteversion methods could be calculated using the WM method (Figure 5).

## 4. Discussion

In this study, we developed 3D models of plain CL radiographs to measure the WM anteversion angle, which can be used to evaluate the actual anteversion of the acetabular component after THA. The schematic figure (Figure 4) provides an idea for readers to understand the optimal position of the acetabular component on a plain CL radiograph. Although virtually reconstructed, it exhibits the reproducibility of an actual plain CL radiograph. The coronal, sagittal, and transverse planes were consistently set by the anatomical structure of the pelvis and could produce the actual orientation of the acetabular component. The “safe-zone” of cup positioning at 5–25° anteversion and 30–50° inclination was defined by Lewinnek et al. [14]. Harris et al. stated that the acetabular component should be inserted during surgery, with a 20° anteversion and 30° inclination [33]. Therefore, in the final schematic figure (Figure 4), the green zone of the six figures presented a tolerable and acceptable position of the acetabular component.

The assessment of the anteversion of the acetabular component is possible on both AP and CL plain radiographs [18,19,28,34]. However, because of the metallic density of the acetabular component, which was white in plain radiographs, anteversion assessment using plain AP radiographs has its limitations. Thus, plain CL radiographs can provide an anteversion assessment of general component positions [23,24,27,28].

The WM method of anteversion can be easily measured and shows good interobserver reproducibility. Therefore, this study developed an intuitive figure to help surgeons determine the optimal acetabular cup position after THA. In addition, this study showed a strong significant positive correlation between the three anteversion measurement methods and the WM anteversion method. By analyzing the WM method of anteversion in a CL plain radiograph, we can calculate the angle of the three original anteversions, which can act as an indicator for the “safe zone” of cup positioning.

The WM method measurement of anteversion in a plain CL radiograph increased as the inclination increased (Figure 5). Therefore, surgeons should understand the correlation between anteversion and inclination. If, in an AP plain radiograph, the inclination of the acetabular cup position is larger than that on a CL plain radiograph, the anteversion of the acetabular cup position should be greater than the estimated value. All three unique anteversions (AA, RA, OA) had high correlation with the Woo and Morrey method. Therefore, we can conclude that with the Woo and Morrey method, we can evaluate the actual anteversion of the hip component after THA and whether it is positioned in “safe zone”.

Acetabular protrusion is rare, and THA is difficult surgery for a better surgical outcome than other osteoarthritis or osteonecrosis of femoral head patients who underwent THA. And an evaluation of anteversion is also difficult. During THA, bone grafting on the medial acetabular wall showed good midterm clinical and radiological results [35]. This was not just for good midterm clinical outcomes; if bone grafting on the medial acetabular wall cannot restore the acetabular component position much medially, the cross-lateral plain radiograph can evaluate more actual anteversion more easily.

This study has some limitations. First, although angle measurement was automatically performed using a computer program, measurement bias remained. As shown in Figure 3, the calculation of anteversion can differ depending on the endpoint setting of the acetabular cup. Second, pelvic tilt can influence anteversion and inclination [36]. As the pelvic tilt decreases, anteversion decreases. Hence, we set all cases in the coronal plane to reflect the pelvic tilt. However, in practice, the pelvic tilt can affect the results. Third, the position of the pelvis differs in each person; therefore, it is difficult to set a standard, universally applicable position in a plain CL radiograph in actual practice. Thus, the standardization of the patient’s position while obtaining a plain radiograph is important. Finally, this study focused mainly on the anteversion and inclination of acetabular components after THA, which can be adjustable when surgeons focus on it and evaluate after surgery. Future studies can focus on patient-related factors and other surgical risk factors.

## 5. Conclusions

The actual anteversion of the acetabular component after THA can be measured on a plain CL radiograph using the WM method of anteversion using a 3D virtual program. Plain CL radiographs showed anteversion of the acetabular cup position with good reproducibility. All three anteversion measurement methods (AA, RA, and OA) showed strong significant positive correlations with the WM method of measurement. Radiographic measurements best matched the WM method, followed by anatomical and operative measurement methods.

## Figures and Tables

**Figure 1 jcm-12-06664-f001:**
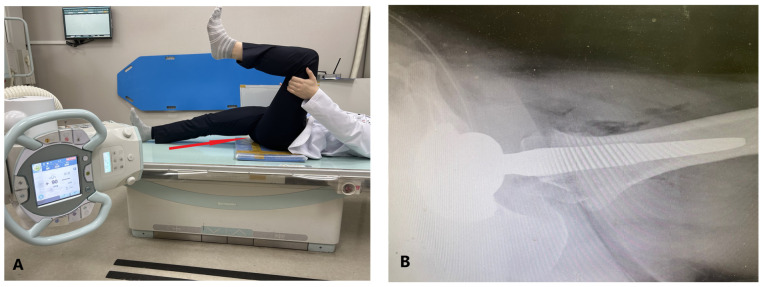
Photograph of the positioning of patients when taking a CL plain radiograph (**A**), and an actual CL plain radiograph (**B**).

**Figure 2 jcm-12-06664-f002:**
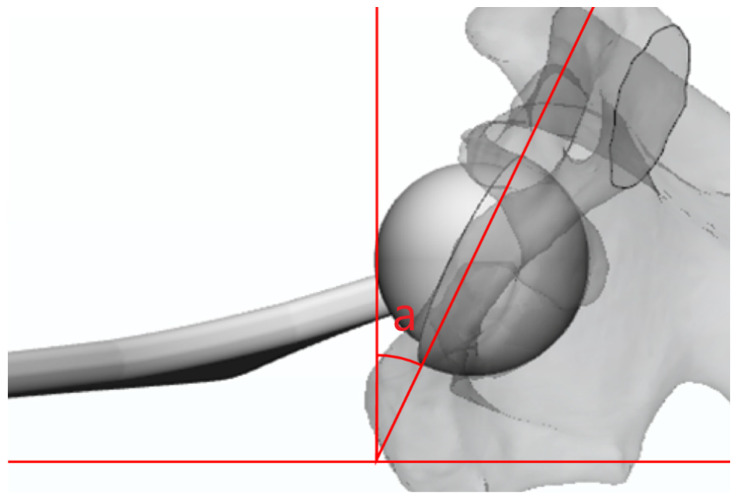
Represented CL plain radiograph on Mimics/calculating acetabular anteversion (a) using the WM method of anteversion on the represented CL plain radiograph on Mimics (red line). CL: cross lateral.

**Figure 3 jcm-12-06664-f003:**
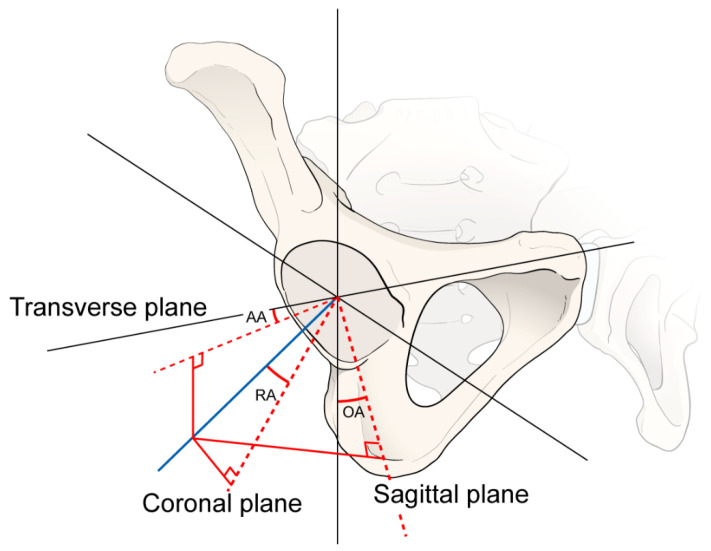
Orientation of the three methods of anteversion and inclination. AA: anatomical anteversion, RA: radiological anteversion, OA: operational anteversion, AI: anatomical inclination, RI: radiological inclination, OI: operational inclination.

**Figure 4 jcm-12-06664-f004:**
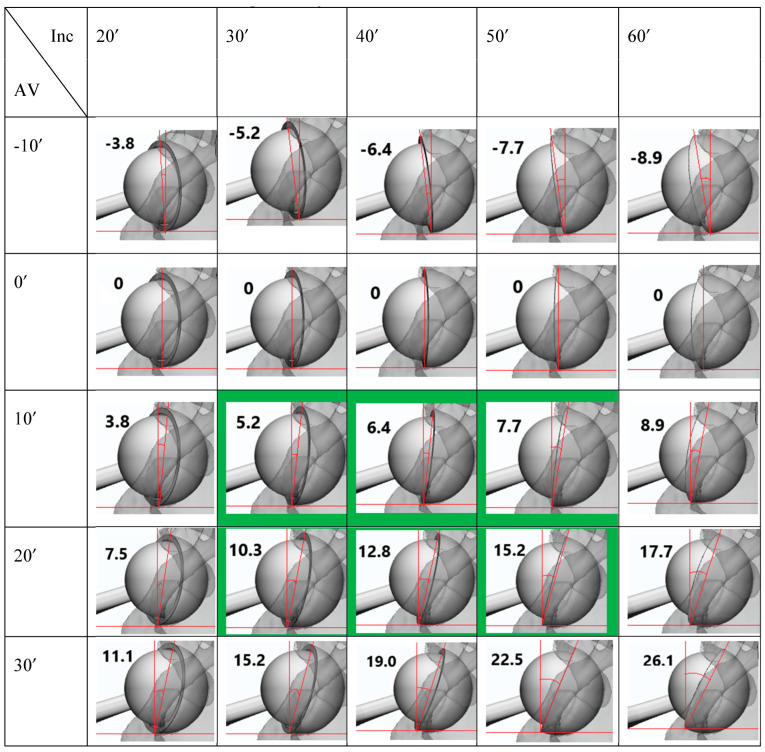
Schematic figure of the five types of anteversion (−10°: retroversion 10°) and the five types of inclination in a CL plain radiograph (number: WM method of anteversion, green zone: safe zone of acetabular component by Lewinnek et al. [14]). If the anteversion is retroverted, the WM method of anteversion is also retroverted. (First row, AV −10°) If anteversion is 0°, the WM method of anteversion measurement is 0°. (Second row, AV 0°) The WM method of anteversion measurement increases as the anteversion and inclination increases. (Third–fifth row, AV 10°, 20°, 30°).

**Figure 5 jcm-12-06664-f005:**
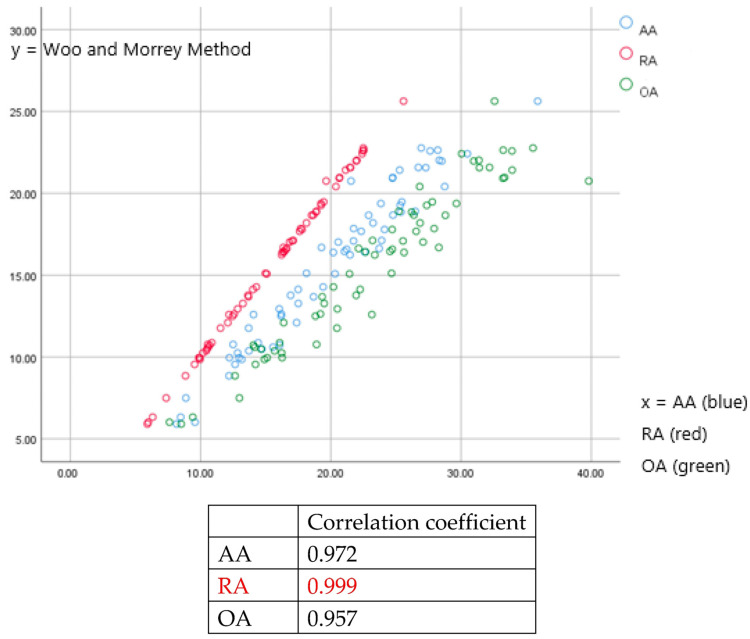
Correlation between the WM method of anteversion and the three methods of anteversion/correlation coefficient of AA, RA, and OA with the WM method of anteversion. The WM method of anteversion shows the most correlation with RA, with a correlation coefficient of 0.999, followed by AA with a correlation coefficient of 0.972, and OA with a correlation coefficient of 0.957.

**Table 1 jcm-12-06664-t001:** Anteversions and inclinations with different measurements and gender distribution.

		Gender		
	Total	Male	Female	*p* Value
AA	20.1 ± 6.1	18.0 ± 5.7	22.6 ± 5.8	0.004
RA	15.5 ± 4.8	14.0 ± 4.6	17.1 ± 4.6	0.008
OA	23.3 ± 7.3	21.4 ± 6.9	25.2 ± 7.2	0.04
AI	51.5 ± 4.7	51.8 ± 5.0	51.1 ± 4.6	0.57
RI	49.5 ± 4.9	50.3 ± 5.0	48.8 ± 4.7	0.23
OI	46.9 ± 4.5	48.0 ± 4.7	45.7 ± 4.1	0.04
WM	15.6 ± 4.8	14.1 ± 4.6	17.2 ± 4.6	0.009

AA: anatomical anteversion, RA: radiological anteversion, OA: operational anteversion, AI: anatomical inclination, RI: radiological inclination, OI: operational inclination, WM: Woo and Morrey method of anteversion.

## Data Availability

The data presented in this study are available on request from the corresponding author.

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
