# Peer review of "Could We Expect Postoperative Cup Anteversion after Total Hip Arthroplasty Using Postoperative Plain Anteroposterior and Lateral Radiograph? A Three-Dimensional Experimental Operation Study"

_jcm, 2023, doi:10.3390/jcm12206664_

Round 1

Reviewer 1 Report

Abstract

- text says "differed significantly" and mentions correlations, but there's no information about the statistical tests used or the level of significance.

- The study does not mention a control group or any comparator, making it hard to assess the effectiveness or reliability of the proposed methods

Manuscript

- Some statements like "it is fatal and requires revision surgery" and "positioning of the acetabular component can be adjusted to reduce the number of dislocations" lack elaboration/supporting data

- please refer to 10.1007/s00264-015-2804-9 and discuss in the background/discussions wether bone grafting would impact the measurements or not, as they are frequently used in uncemented hip replacement. 

- The background cites some statistics and prior research but seems to focus heavily on issues related to dislocation. It doesn't provide a broader view of other complications or issues in THA that might be relevant

- authors notes that "the terms 'inclination and anteversion' are frequently confused with these three methods," but doesn't clarify what those terms mean or how they are commonly confused. A definition or clarification here would help set the stage for the study.

- The text specifies exclusion criteria but does not give detailed inclusion criteria for the study, other than the patients having undergone THA.

- There is no mention of how the sample size of 32 patients was determined to be statistically sufficient for the study

- the aim proposes to determine "ideal" angles but doesn't explain how these ideals are validated or why the specific angles chosen are considered to represent a range of "ideal" positions

- While the authors mentions analyzing the hip acetabular anatomy to calculate various angles, it's not clear how this contributes to the primary objective of determining the ideal acetabular component position in THA?

Author Response

Abstract

- text says "differed significantly" and mentions correlations, but there's no information about the statistical tests used or the level of significance.

Thank you for your valuable comment. We added p value < 0.05 (line  23), which is significantly different.

- The study does not mention a control group or any comparator, making it hard to assess the effectiveness or reliability of the proposed methods

In table 1, there are two groups for gender distribution. Because the anteversion and inclination of one person is the the person’s unique angle, statistical analysis can be done on 64 hips. The virtual 3D surgery was done on one sample pelvis for different anteversion and inclination of acetabular component, so there is no statistical analysis on figure 4.

However, there is correlation coefficient between AA, RA, OA with Woo and Morrey method.(figure 5) All three unique anteversion (AA, RA, OA) of patient have high correlation with Woo and Morrey method. Moreover, Woo and Morrey method is represented in cross-lateral plain radiograph, we can evaluate anteversion after THA through cross-lateral plain radiograph.

We fully agreed the message is not sufficiently discussed and presented in manuscript, so we revised and added it. (line 224-227)

Manuscript

- Some statements like "it is fatal and requires revision surgery" and "positioning of the acetabular component can be adjusted to reduce the number of dislocations" lack elaboration/supporting data

We agreed the statements need to be revised. So we repharsed it and added references. (line 39-40, reference #10, #11) (line 47-49, reference #13)

- please refer to 10.1007/s00264-015-2804-9 and discuss in the background/discussions wether bone grafting would impact the measurements or not, as they are frequently used in uncemented hip replacement. 

Thank you for your comment. We added the reference and added the paragraph in discussion section. (line #228-234, reference #37)

- The background cites some statistics and prior research but seems to focus heavily on issues related to dislocation. It doesn't provide a broader view of other complications or issues in THA that might be relevant

This study focused mainly on the anteversion and inclination of acetabular component after THA, which can be adjustable when surgeons focus on it and evaluate after surgery. We would like to give radiological insight for novice hip surgeons for how much angle on postoperative x-ray is tolerable range of anteversion of acetabular component after THA. We agreed that we didn’t focus on other factors of patient-related and surgical risk factors in broader view of other complications. Future studies can focus on patient-related factors and other surgical risk factors. We added it in limitation section. (line #244-247)

- authors notes that "the terms 'inclination and anteversion' are frequently confused with these three methods," but doesn't clarify what those terms mean or how they are commonly confused. A definition or clarification here would help set the stage for the study.

We agreed on it and added more explanations of it.(line #72-75) And the exact definition of it is already written in method section (line #122-132) with figure 3.

- The text specifies exclusion criteria but does not give detailed inclusion criteria for the study, other than the patients having undergone THA.

We added it. (line #93-94) The data was the pre operative CT scan of patients who underwent primary THA so as to obtain relatively clear image of CT scan.

- There is no mention of how the sample size of 32 patients was determined to be statistically sufficient for the study

We added it in “statistical analysis”, line 148-150 with references.(reference #32, #33)

- the aim proposes to determine "ideal" angles but doesn't explain how these ideals are validated or why the specific angles chosen are considered to represent a range of "ideal" positions

We agreed that “ideal” should be defined. We based on the idea of “safe zone” of Lewinnek et al. cup positioning: 5–25° of anteversion and 30–50° of inclination. So we revised the figure 4 and defined the term acceptable position to safe zone. (line #14) (line #20) (line #100-101) (line #134-136) (line #140)

- While the authors mentions analyzing the hip acetabular anatomy to calculate various angles, it's not clear how this contributes to the primary objective of determining the ideal acetabular component position in THA?

We agreed on that point and revised all terms clearly. There is no absolute cup position of acetabulum, only there is safe zone of cup positioning. So we revised the term. (line #14) (line #20) (line #100-101) (line #134-136) (line #140)

Reviewer 2 Report

The aim is to determine the exact anteversion of the acetabular component on postoperative radiographs by obtaining correlation data between virtual and actual acetabular component positioning using three-dimensional (3D) virtual surgery.

Major: Correlation and prediction are different concepts. Edit the method and term.

 There are many biases, such as measurement bias.

Minor

Outcome: Does the exact anteversion of the acetabular component on postoperative radiographs mean Radiographic anteversion (RA) and Radiographic inclination (RI)? Very unclear ..

L138: Statistical analysis, calculate the sample size. 64 hips is enough?

L138: Statistical analysis,

After completing the graph of correlation, please indicate the range of error between the angle calculated by the graph and the actual angle.

L140:To examine the correlation of the three methods of anteversion and inclination measurement, ICC may be wrong. It is common to use correlation coefficients.

Fig 5: describe the name of the X and Y axes.

Author Response

The aim is to determine the exact anteversion of the acetabular component on postoperative radiographs by obtaining correlation data between virtual and actual acetabular component positioning using three-dimensional (3D) virtual surgery.

Major: Correlation and prediction are different concepts. Edit the method and term.

Thank you for your valuable comment. We agreed that revised it. (line #189) We do not use the term ”prediction” for unified concept of this article.

 There are many biases, such as measurement bias.

We agreed with it. We added it in limitation section (line #235-238)

Minor

Outcome: Does the exact anteversion of the acetabular component on postoperative radiographs mean Radiographic anteversion (RA) and Radiographic inclination (RI)? Very unclear ..

The message is that Woo and Morrey method of anteversion has strong correlation with radiograghic anteversion. And the inclination of acetabular component can be easily measured in AP plain radiograph, which is the radiographic inclination. So for surgeons, inclination is quite easily evaluated via AP x-ray, but anteversion is not easily evaluated on plain radiograph. We agreed on your point and revised some term “exact” to “actual” (line #9, line #83-84, line #194)

L138: Statistical analysis, calculate the sample size. 64 hips is enough?

We added it in “statistical analysis”, line 148-150 with references.(reference #32, #33)

L138: Statistical analysis,

After completing the graph of correlation, please indicate the range of error between the angle calculated by the graph and the actual angle.

The AA, RA, OA is unique angle which is determined by patient’s on hip. And Woo and Morrey method of anteversion is the angle that we measured in virtual cross-lateral plain radiograph. So there can be only correlation coefficient between three angles and WM angle.

L140:To examine the correlation of the three methods of anteversion and inclination measurement, ICC may be wrong. It is common to use correlation coefficients.

We agreed on your point so we deleted all ICC method. Only correlation coefficients still remained.

Fig 5: describe the name of the X and Y axes.

Thank you for your comment. We revised it.

Round 2

Reviewer 1 Report

Authors have prepared the manuscript with a better form and it can now be published.

Reviewer 2 Report

No comments